# Intentional Replantation as a Starting Approach for a Multidisciplinary Treatment of a Mandibular Second Molar: A Case Report

**DOI:** 10.3390/jcm11175111

**Published:** 2022-08-30

**Authors:** João Miguel Santos, Joana A. Marques, Margarida Esteves, Vítor Sousa, Paulo J. Palma, Sérgio Matos

**Affiliations:** 1Institute of Endodontics, Faculty of Medicine, University of Coimbra, 3000-075 Coimbra, Portugal; 2Center for Innovation and Research in Oral Sciences (CIROS), Faculty of Medicine, University of Coimbra, 3000-075 Coimbra, Portugal; 3Coimbra Institute for Clinical and Biomedical Research (iCBR) and Center of Investigation on Environment Genetics and Oncobiology (CIMAGO), Faculty of Medicine and Clinical Academic Center of Coimbra (CACC), 3000-548 Coimbra, Portugal; 4Institute of Paediatric and Preventive Dentistry, Faculty of Medicine, University of Coimbra, 3000-075 Coimbra, Portugal; 5Institute of Pathological Anatomy, Faculty of Medicine, University of Coimbra, 3000-548 Coimbra, Portugal; 6Institute of Periodontology, Faculty of Medicine, University of Coimbra, 3000-075 Coimbra, Portugal

**Keywords:** alveolar ridge augmentation, Biodentine, intentional replantation, leukocyte-platelet rich fibrin, post-treatment apical periodontitis, pulp vitality

## Abstract

Intentional replantation (IR) may offer a solution for persistent periapical lesions associated with endodontically treated teeth. A 35-year-old male patient presented with pain associated with the left mandibular second molar and hypoesthesia. Upon clinical examination, increased probing pocket depth in the mid-buccal surface was detected. Cone beam computed tomography revealed a previous non-surgical root canal treatment, with root canal filling material extrusion adjacent to the inferior alveolar nerve, a fractured instrument in the mesial root, and a large periapical radiolucency involving both teeth 37 and 36. A diagnosis of symptomatic post-treatment apical periodontitis was established. After discussing treatment options with the patient, an IR of tooth 37 was performed. Extra-oral procedures were completed in 17 min. At 9 months, hypoesthesia resolution was reported, and apical healing was radiographically observed. After 2.5 years, the replanted tooth showed extensive root resorption. An extraction with alveolar ridge preservation, using leukocyte-platelet rich fibrin (L-PRF), was performed. Six months after tooth extraction and regeneration, implant placement surgery was carried out. IR presents a valid treatment modality for the management of post-treatment apical periodontitis. When orthograde retreatment or apical microsurgery prove to be unfeasible, IR is a unique procedure with the potential to promote tooth preservation in properly selected cases. Although unsuccessful after 2.5 years, the IR of tooth 37 allowed for bone regeneration, the maintenance of tooth 36 vitality, and hypoesthesia resolution.

## 1. Introduction

Intentional replantation (IR) is a conservative and economical treatment alternative for the management of post-treatment apical periodontitis when orthograde retreatment or apical microsurgery is unfeasible, has failed, or is linked with risks that are unacceptable for the patient [1,2,3].

IR consists of the deliberate extraction of the affected tooth and its reinsertion into the alveolus after sequential root resection, preparation, and hermetic sealing with a biocompatible root-end filling material [4]. The ultimate goal of this procedure is to enable tooth survival [5].

The IR technique has been progressively modified and refined. Nowadays, it involves atraumatic tooth extraction methods, root resection, and preparation using piezoelectric systems; extra-oral handling for the shortest possible period in an environment suitable for maintaining cell viability; and root-end filling with different biomaterials, performed under illumination and magnification [6,7,8]. A survival rate of 88% at 2 years has been reported [9]. However, several pre- and perioperative factors may significantly impact prognosis [8,10,11]. When performed in accordance with contemporary apical microsurgery knowledge and the biological basis of dental trauma, this technique maximizes healing potential and reduces the risk of developing complications [1,6]. Ankylosis, root resorption, the expansion of apical radiolucency, the persistence or intensification of symptoms, and increased probing depth are the main complications described within the literature [12].

This case report describes the multidisciplinary management of a mandibular second molar through IR with 2.5 years of follow-up, followed by extraction with alveolar ridge preservation using L-PRF and implant-supported prosthetic rehabilitation.

## 2. Case Report

A male patient, who was 35 years old, Caucasian, and healthy, referred pain associated with the left mandibular second molar (37) while chewing and hypoesthesia. The previous medical history included a non-surgical root canal treatment (NSRCT), performed across multiple sessions, that also involved an instrument fracture. The tooth remained asymptomatic for approximately 10 years after the NSRCT. Upon clinical examination, a distal coronal vertical craze line and increased probing pocket depth in the mid-buccal surface (9 mm) were detected (Figure 1a,b). Cone beam computed tomography (CBCT) revealed the previous non-surgical root canal treatment, along with root canal filling material extrusion adjacent to the inferior alveolar nerve, a fractured instrument in the apical third of the mesial root, and a large periapical radiolucency involving both teeth 37 and 36 (Figure 1(c1,c2)). Sensitivity and electric pulp tests were positive for tooth 36. Additionally, a photographic record was taken (Canon EOS 70D camera, EF 100 mm 1:2.8 L IS USM Macro Lens and Macro Ring Lite MR-14EXII, F29 aperture, ISO 100, 1/125 shutter speed and flash at ETTL mode, −1⁄3 power). A pulp diagnosis of previously treated tooth and a periapical diagnosis of symptomatic apical periodontitis were established. After discussing treatment options (orthograde retreatment, apical microsurgery, IR, and extraction with proper alveolus management and delayed implant placement) and associated risks and benefits with the patient, a decision arose to perform an IR of tooth 37, and informed consent was obtained.

### 2.1. Intentional Replantation

Preoperative antisepsis of the oral cavity (0.2% chlorhexidine digluconate solution rinse for 1 min) and infiltrative anesthesia (2% lidocaine with 1:80.000 epinephrine) were carried out. After separating the periodontal fibers through an intrasulcular incision parallel to the tooth’s long axis with a 12-blade surgical scalpel (Figure 1d), atraumatic extraction by exerting a controlled force with dental forceps was carried out (Figure 1e). Extra-oral handling started with meticulous root surface inspection (Figure 1f) and granulation tissue removal (Figure 1g) while wrapping the tooth’s crown in saline-soaked gauze. An apicoectomy (3 mm), perpendicular to the tooth’s long axis, was performed with a high-speed truncated-conical diamond bur (FG ML 200651AA, ISO 290012, Diatech; Coltène/Whaledent, Altstätten, Switzerland) under continuous water spray (Figure 1h), followed by 3 mm deep apical preparation using ultrasonic tips (E32D; NSK, Tochigi, Japan) (Figure 1i), with the removal of the fractured instrument (Figure 1j). Subsequently, the apical cavity was dried and filled with Biodentine (Septodont, Saint-Maur-des-Fossés Cedex, France) (Figure 1k,l). Extra-oral procedures were completed in 17 min under 10× magnification (Leica Microscope M320; Leica Microsystems, Heerbrugg, Switzerland). While tooth extraoral handling procedures were being performed, the alveolus was prepared under 6× magnification. Alveolus preparation included careful and meticulous curettage, using a surgical curette for periapical lesions and extruded filling material removal, in alternation with thorough sterile saline solution rinsing. The tooth was then replanted in the previously prepared alveolus (Figure 1m) and the patient was instructed to bite on gauze. No splinting was required, as primary stability was confirmed. Both extra-oral handling and alveolus preparation procedures were performed by two experienced operators (J.M.S. and P.J.P., respectively) in order to reduce operative time. An anatomopathological analysis of the removed periapical lesion identified a chronic inflammatory process associated with granulation tissue formation, abundant inflammatory infiltrates, and exogenous material (Figure 1n,o). A reduction in the mid-buccal surface probing pocket depth to 5 mm, with no bleeding on probing, was verified at the 2-month postoperative control. At 9 months, hypoesthesia resolution was reported by the patient. Moreover, although apical healing with bone architecture reestablishment was radiographically observed (Figure 2a), periodontal inflammation was found in the mid-buccal region. Mechanical debridement with ultrasound tips (P20; NSK, Tochigi, Japan), topical application of minocycline and chlorhexidine gel, adjustment of occlusal contacts, and antibiotic prescription (amoxicillin and clavulanic acid 875/125 mg given every 12 h for 8 days) were performed. After 10 months, given the persistence of the periodontal buccal inflammation, new subgingival instrumentation was performed through mechanical ultrasonic debridement, as well as glycine powder air-polishing and additional topical applications of minocycline and chlorhexidine gel. At the 1-year follow-up, the probing pocket depth was 3 mm, with no clinical signs of inflammation. Twenty months after IR, a mesiobuccal suppurative pocket (6 mm depth) was found. A CBCT showed overall bone regeneration, although it also confirmed the existence of a residual mesial infrabony defect (Figure 2(b1,b2)). A third mechanical ultrasonic debridement was performed, followed by glycine powder air-polishing. At 2.5 years, the radiographic examination of the replanted tooth showed extensive external root resorption [13]. Thus, clinical indication for tooth extraction was established (Figure 2c).

### 2.2. Extraction with Alveolar Ridge Preservation Using Leukocyte-Platelet Rich Fibrin (L-PRF)

Before tooth extraction, L-PRF preparation was performed according to the manufacturer (IntraSpin^TM^ centrifuge, Intra-Lock, Boca Raton, FL, USA) and the established clinical protocol [14,15]. After blood collection, L-PRF membranes and plugs were prepared (Figure 3a–e). The atraumatic tooth extraction and meticulous curettage of the alveolus were performed, and an absence of most of the coronal buccal bone plate was observed (Figure 3f–j). A small full-thickness envelope tunnel was achieved surrounding the socket (2 mm wide) to immobilize L-PRF membranes between the periosteum and the flap. After copious alveolus irrigation with L-PRF exudate, alveolar ridge preservation was performed by applying three plugs and five L-PRF membranes (including two double membranes), followed by a midbuccal modified internal mattress suture and double sling sutures on the base of the papilla (5/0 monofilament suture; Seralon^®^, Serag Wissner^™^, Naila, Germany), with the intention of keeping the membranes in place without traction (Figure 3k–q). All previously described procedures were implemented by two experienced operators (J.M.S. and S.M.). Reassessments were performed at 7 days (Figure 3r), 1 month (Figure 3s), and 5 months (Figure 3t).

### 2.3. Implant Surgery

Six months after tooth extraction and regeneration (Figure 4a), the implant placement surgery was performed by one experienced operator (J.M.S.). After mucoperiosteal flap elevation, a reconstructed bony crest was observed with a small residual defect on the mesial aspect but with an integrated buccal wall. Site preparation and implant placement (OsseoSpeed^TM^ EV 5.4 S–11 mm–Ref.26363; Dentsply Sirona, Hanau, Germany) were carried out (Figure 4b–d). Thereafter, the healing abutment (HealDesign^TM^ EV 5.4–6.5 mm–Ref.25799; Dentsply Sirona, Hanau, Germany) was positioned, followed by suturing with 4/0 silk (Silkam; B.Braun Surgical, Rubí, Spain) (Figure 4e). After 3 months, implant impressions were taken using an Implant Pick-Up EV (Ref.26230; Dentsply Sirona, Hanau, Germany) (Figure 4f–h). Prosthetic rehabilitation through screw-retained implant zirconia crown placement was accomplished (Figure 4i,j). Follow-ups at 3, 6, and 12 months (Figure 4k) after implant surgery were carried out. Thermal and electrical tests remained positive for tooth 36 at the last follow-up appointment.

## 3. Discussion

This case report describes the management of a mandibular second molar that underwent root resorption 2.5 years after IR and then further submitted to extraction with alveolar ridge preservation using L-PRF and implant-supported prosthetic rehabilitation.

Following the patient’s medical history, clinical, and radiographic examinations, the clinical decision-making process implied discussion of all treatment options with the patient: (1) orthograde retreatment, (2) apical microsurgery, (3) IR, or (4) extraction with proper alveolus management and delayed implant placement. The rationale for not proceeding with orthograde retreatment was related to a combination of multiple preoperative factors that undermine the success of this approach—namely, the presence of a large periapical lesion involving the apical region of both teeth 37 and 36, the existence of root canal filling material close to the inferior alveolar nerve, as well as a fractured instrument in the apical third of the mesial root of tooth 37. On the other hand, apical microsurgery presented an alternative option with risks associated with the anatomical proximity of the inferior alveolar nerve and technical difficulties arising from the retropreparation and root-end filling of a mandibular second molar with a dense and thick buccal cortical bone. Considering the limitations of the aforementioned approaches, whenever they are determined to be unfeasible, IR presents a conservative, favorable, and cost-effective last resort option over tooth extraction and prosthetic rehabilitation, with the potential to promote the preservation of the natural tooth and consequent bone tissue maintenance or improvement [2,3,6,11]. IR incorporates the benefits of orthograde retreatment and apical microsurgery by allowing for the control of both intra- and extra-radicular, as well as periapical, infections. Contrary to apical microsurgery, IR does not involve soft tissue handling, thus presenting a less invasive procedure and possessing a lower chance of marginal bone loss or dehiscence. Further, it becomes more predictable to achieve hermetic and effective apical sealing as the root-end filling is extra-orally performed [2].

However, several pre- and perioperative prognostic factors have been identified. The number of periodontal pockets (≥6 mm) and patient age (>40 years old) comprise key pre-operative aspects that may jeopardize the success of the IR procedure [16]. Contrastingly, peri-operative factors enhancing IR success include atraumatic extraction methods, the use of piezoelectric systems for root resection and preparation, cellular and acellular cement preservation, extraoral handling time ≤ 15 min, bioactive root-end filling material, and the use of magnification [6,10,12,17,18,19]. These highlight the importance of performing IR in light of modern microsurgical endodontics principles as requirements for a favorable prognosis [1,6,20]. Aside from microsurgical instruments, ultrasonic tips, and magnification, contemporary techniques include the use of biocompatible root-end filling materials, such as Mineral Trioxide Aggregate and Biodentine [6,10,18,21,22,23]. Moreover, fast-set, putty-like, premixed, calcium silicate-based cements are currently available, which may help to reduce intervention time [10,24].

A previous study reported that healing occurred 1.7 times more frequently when teeth were replanted within 15 min [12]. During extraoral handling, periodontal ligament cells experience an interruption in blood supply and dehydration, with long extraoral periods leading to cell viability decreases and periodontal healing impairment [6,10,19]. In fact, extraoral working times longer than 15 min have been associated with a higher incidence of postoperative complications, thus underlining the need for adequate surgical planning [2,10,12]. Although IR became a widely recommended approach for the management of specific clinical scenarios, proper case selection based on the abovementioned factors is crucial for a successful prognosis [11]. Teeth presenting with divergent or curved roots, fractured coronal or radicular structures, periodontal involvement with associated mobility, as well as compromised vestibular and/or lingual/palatal bone plates, may indicate contraindications for performing IR [2,4,11]. In this case report, since all the remaining factors were addressed, the authors speculate that the presence of a pre-operative 9 mm periodontal pocket and a 17-min extraoral working time (even though two experienced operators were assured) can be pinpointed as the reasons for IR failure after 2.5 years due to extensive root resorption. As previously referred to, root resorption presents one of the primary adverse events following IR [9]. According to Cho et al. [12], most complications take place during the first year after IR, with a small increase in periapical radiolucency and external root resorption occurrence between the first and third postoperative years. Therefore, a minimum 3-year follow-up period has been suggested. Alternative treatment options could involve guided eruption techniques, as orthodontic extrusion 2 to 3 weeks prior to tooth extraction holds the potential for preventing intra- and post-operative IR complications [8]. This therapeutic approach can also be used for tissue regeneration and implant site development using either lingual or conventional mechanics [25].

In 2015, by comparing the survival of intentionally replanted teeth reported across 8 articles with the survival of implant-supported single crowns reported across 27 articles, Torabinajed et al. outlined an 88% survival rate of replanted teeth at two years [9]. In 2016, Cho et al. carried out a prospective study with 159 patients, mostly females under 40 years old, in which the majority of the intentionally replanted teeth were second molars with apical periodontitis, adequate root canal filling, and the absence of a fistula. Most teeth were root-end filled with intermediate restorative material and replanted in less than 15 min. In this study, a cumulative retention rate of 93% at 12 years and a cumulative rate of clinical and radiographic healing of 91% at 6 months, which decreases to 77% at 3 years, were described [12]. In 2017, Mainkar conducted a meta-analysis that revealed higher survival rates for single implants over IR. However, IR emerged as a more cost-effective treatment option [1].

Despite the high survival rates found within the literature, unsuccessful IR does not jeopardize, and can even improve, the conditions of hard and soft tissues for subsequent extraction and prosthetic rehabilitation, thus exhibiting a valid treatment modality that should be discussed with the patient [1]. In the present case report, notwithstanding the radiographic evidence of bone regeneration seen in Figure 2(b1,b2) when compared to the initial situation (Figure 1(c1,c2)), the ensuing development of extensive root resorption led to bone support loss and infection. Therefore, post-extraction alveolar ridge preservation, using L-PRF, was performed in order to optimize hard tissue conditions for the subsequent placement of a single implant. L-PRF, more than a biomaterial, can be considered a vital tissue precursor for tissue engineering. From the extracted blood, 97% of the platelets and more than 50% of the leukocytes are concentrated inside the PRF clot. After its application for approximately 2 weeks, growth factors such as PDGF, TGF-B1, VEGF, and BMP-1 are released by platelets, which induce and control the proliferation and migration of regenerative cells, namely progenitor and stem cells. In addition, it acts as an immunomodulator and inflammation regulator [26,27]. The use of L-PRF as a socket filling material to achieve the preservation of horizontal and vertical ridge dimensions has been reported in the literature with beneficial clinical outcomes, and significant improvements have been disclosed compared with empty biomaterial controls with a single stabilization of the coagulum [25,26]. This approach, despite reducing alveolar resorption, is associated with less postoperative discomfort for the patient and allows for an optimized reentry time for implant placement [27,28,29,30,31]. In addition to bone condition enhancement, it is essential to emphasize that the IR approach to the large preoperative periapical lesion that involved both the mesial and distal roots of tooth 36 enabled the maintenance of its vitality, ultimately preventing the need for endodontic treatment [32]. Moreover, hypoesthesia resolution following IR must be highlighted as a valuable patient-centered outcome.

Regarding IR limitations, the inexistence of an established clinical protocol, as well as the heterogeneity of success criteria, translates into the scarcity of standardized studies and the reporting of a wide range of success and survival rates [6,9]. Furthermore, the scarcity of recent studies focusing on IR, along with the vast available literature on orthograde retreatment and apical microsurgery, also becomes a barrier to the clinical application of this procedure. Lastly, it is difficult to anticipate intraoperative difficulties for achieving atraumatic tooth extraction and performing all extraoral procedures within 15 min, which may discourage clinicians from proposing IR to the patient.

## 4. Conclusions

Overall, it can be concluded that IR presents a valid treatment modality for the management of post-treatment apical periodontitis. When orthograde retreatment or apical microsurgery prove to be unfeasible, IR is a unique procedure with the potential to promote tooth preservation in properly selected cases. Although unsuccessful after 2.5 years, IR allowed for bone regeneration, the maintenance of tooth 36 vitality, and hypoesthesia resolution. In the case of IR failure, extraction with alveolar ridge preservation using L-PRF and subsequent prosthetic rehabilitation through implant placement is a predictable therapy approach.

## Figures and Tables

**Figure 1 jcm-11-05111-f001:**
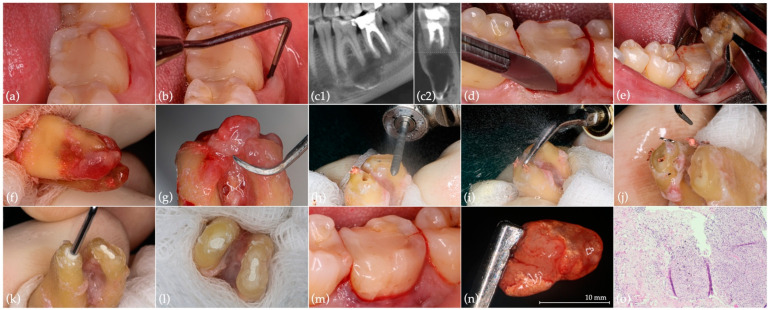
Preoperative situation and IR of tooth 37: (**a**) initial situation; (**b**) mid-buccal probing pocket depth (9 mm); (**c1**) sagittal and (**c2**) coronal CBCT section of the region of interest; (**d**) periodontal fiber separation; (**e**) atraumatic extraction with dental forceps; (**f**) meticulous inspection of the root surface; (**g**) granulation tissue removal using periodontal scalers; (**h**) root resection; (**i**) root-end ultrasonic preparation; (**j**) fractured instrument removal; (**k**) root-end filling with Biodentine; (**l**) final aspect of the root-filled apical cavities, with evident isthmus in the mesial root; (**m**) tooth replanted in the previously prepared alveolus (by a thorough sterile saline solution rinsing and periapical lesion and extruded filling material removal), without splinting; (**n**) removed large periapical lesion; (**o**) histological section of the removed periapical lesion showing granulation tissue with abundant inflammatory infiltrate and exogenous material (hematoxylin and eosin staining, 200× magnification).

**Figure 2 jcm-11-05111-f002:**
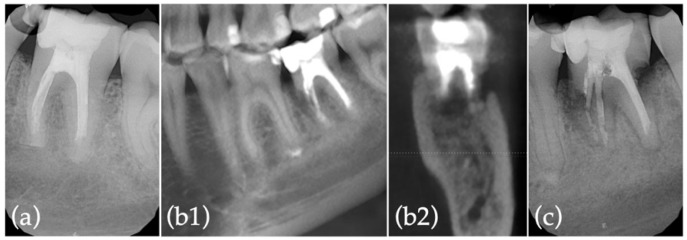
Radiographic records of IR postoperative controls: (**a**) 9-month follow-up; (**b1**) sagittal and (**b2**) coronal CBCT sections of the region of interest at 20-month follow-up; (**c**) 2.5-year follow-up.

**Figure 3 jcm-11-05111-f003:**
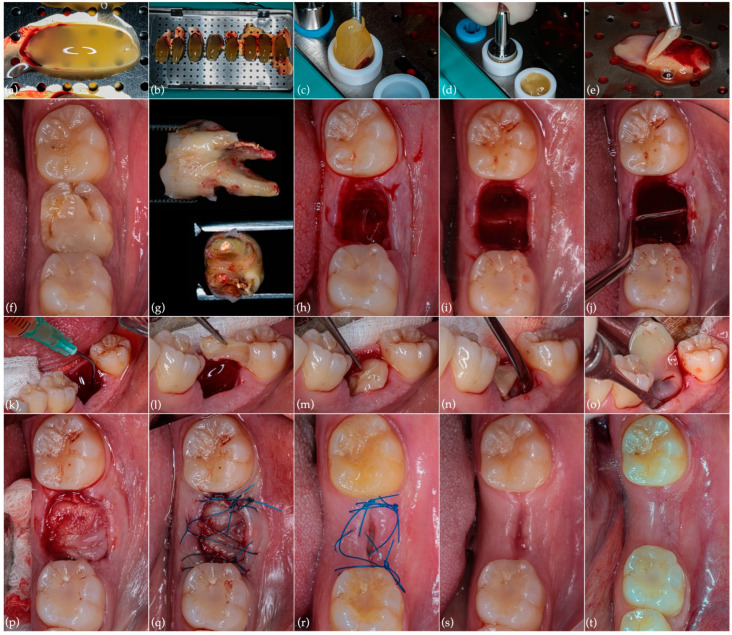
Extraction and alveolar ridge preservation procedures and postoperative controls: (**a**,**b**) L-PRF membranes and (**c**,**d**) plug preparation; (**e**) double L-PRF membrane; (**f**) preoperative occlusal aspect; (**g**) extracted tooth exhibiting extensive external root resorption; (**h**) immediate post-extraction alveolus; (**i**) post-extraction alveolus following meticulous curettage; (**j**) absence of buccal bone plate; (**k**) copious alveolus irrigation with blood plasma; (**l**,**m**) alveolar ridge preservation through the application of three L-PRF plugs and (**n**,**o**) five membranes immobilized in the surrounding tunnel; (**p**) aspect after completion of L-PRF procedures; (**q**) suture with 5/0 monofilament; (**r**) 7-day follow-up; (**s**) 1-month follow-up; (**t**) 5-month follow-up.

**Figure 4 jcm-11-05111-f004:**
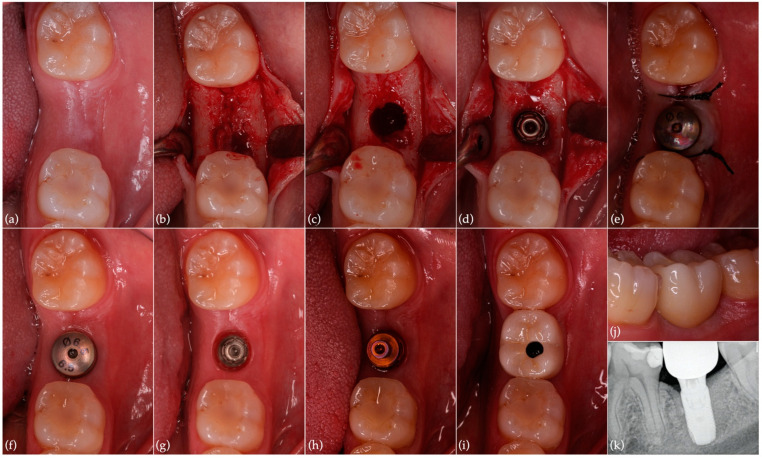
Implant surgery, prosthetic rehabilitation and last postoperative control: (**a**) clinical aspect 6 months after tooth extraction with alveolar preservation; (**b**) elevation of mucoperiosteal flap; (**c**) implant site preparation; (**d**) implant in place (OsseoSpeed^TM^ EV 5.4 S–11 mm); (**e**) healing abutment in position and suture; (**f**) three months after surgery; (**g**) implant-soft tissue interface; (**h**) implant pick-up; (**i**,**j**) screw-retained implant zirconia crown in place; (**k**) periapical radiograph of the region of interest 1 year after implant surgery.

## Data Availability

Not applicable.

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
