# Peer review of "Intentional Replantation as a Starting Approach for a Multidisciplinary Treatment of a Mandibular Second Molar: A Case Report"

_jcm, 2022, doi:10.3390/jcm11175111_

Round 1
Reviewer 1 Report
The authors presented a case report of intentional replantation as a starting approach for a multidisciplinary treatment of a mandibular second molar. It is also a starting point for a discussion about possible treatment options in terms of overall success rate and long-term tooth prognosis observed in similar cases. The manuscript is well written with minor spell-check required. However there are some small remarks:
- The diagnosis of symptomatic post-treatment apical periodontitis was established - maybe more accurate: chronic exacerbated apical periodontitis. How long ago the tooth was treated before? If there's a lesion it must have been a long time. Was it only a single treatment or multiple conservative approach? A bit more about history of the tooth would be recommended.
- Line 91: a diamond bur under copious irrigation… irrigation is more referred to endo procedure. Rather a water spray is visible on a figure.
- Apicoectomy or apicectomy?
- Shorter sentences are recommended eg: Line 134/135: rewrite. Confusing.
- Amoxicillin with clavulonic acid: the dose and time shall also be added.
- Line 119: through ? or maybe „by a thorough saline solution rinsing”
- Line 205/206: Site preparation and implant placement (OsseoSpeedTM…) were carried out ( Fig..). The order.
- Line 259: buccal cortical bone.
- A small comment about the role of PRF in a healing process would be recommended in the discussion.
Author Response
- The authors presented a case report of intentional replantation as a starting approach for a multidisciplinary treatment of a mandibular second molar. It is also a starting point for a discussion about possible treatment options in terms of overall success rate and long-term tooth prognosis observed in similar cases. The manuscript is well written with minor spell-check required. However there are some small remarks:
Author’s response: We thank the reviewer #1 for the comments.
Revised text: Not applicable.
- The diagnosis of symptomatic post-treatment apical periodontitis was established - maybe more accurate: chronic exacerbated apical periodontitis. How long ago the tooth was treated before? If there's a lesion it must have been a long time. Was it only a single treatment or multiple conservative approach? A bit more about history of the tooth would be recommended.
Author’s response: Following the reviewer #1 valuable suggestion, additional details regarding history of the tooth were added. Moreover, following the Glossary of Endodontic Terms provided by the American Association of Endodontists, pulp and periapical diagnosis nomenclature was updated.
Revised text:
“Previous medical history includes a non-surgical root canal treatment (NSRCT) perfomed in multiple sessions and involving an instrument fracture. The tooth kept asymptomatic for approximately 10 years after NSRCT.” – Case Report section, page 2, lines 70 to 73.
“Pulp diagnosis of previously treated and periapical diagnosis of symptomatic apical periodontitis were established.” – Case Report section, page 2, lines 82 and 83.
- Line 91: a diamond bur under copious irrigation… irrigation is more referred to endo procedure. Rather a water spray is visible on a figure.
Author’s response: Once again we thank the reviewer #1 for the risen point. The manuscript has been altered following the reviewer suggestion.
Revised text: “…under continuous water spray (Figure 1h)…” – Case Report section, page 3, line 222.
- Apicoectomy or apicectomy?
Author’s response: Both terms are synonymous and refer to the root-end resection procedure. However, according to the Glossary of Endodontic Terms provided by the American Association of Endodontists, apicoectomy is in fact the recommended term to use. Therefore, manuscript correction was carried out.
Revised text: “Apicoectomy (3 mm)…” – Case Report section, page 3, line 220.
- Shorter sentences are recommended eg: Line 134/135: rewrite. Confusing.
Author’s response: We thank the reviewer #1 for the comment. The highlighted sentence was rewritten.
Revised text: “At 2.5 years, radiographic examination of the replanted tooth showed extensive external root resorption [13]. Thus, clinical indication for tooth extraction was established (Figure 2c).” – Case Report section, page 4, lines 270 to 272.
- Amoxicillin with clavulanic acid: the dose and time shall also be added.
Author’s response: Following the reviewer’s instructions, details regarding antibiotic dosage were added.
Revised text: “…(amoxicillin and clavulanic acid 875/125 mg given every 12 hours for 8 days)…” – Case Report section, page 4, lines 261 and 262.
- Line 119: through ? or maybe „by a thorough saline solution rinsing”.
Author’s response: Following the reviewer’s suggestion, sentence was rewritten.
Revised text: “While tooth’s extraoral handling procedures were being performed, the alveolus was prepared under 6x magnification. Alveolus preparation included a careful and meticulous curettage using a surgical curette for periapical lesion and extruded filling material removal, alternating with thorough sterile saline solution rinsing. The tooth was then replanted in the previously prepared alveolus (Figure 1m) and patient was instructed to bite on gauze.” – Case Report section, page 3, lines 228 to 231.
- Line 205/206: Site preparation and implant placement (OsseoSpeedTM…) were carried out ( Fig..). The order.
Author’s response: Corrected.
Revised text: “…OsseoSpeedTM EV 5.4 S - 11 mm…” - Case Report section, page 5, lines 353 and 354.
- Line 259: buccal cortical bone.
Author’s response: We thank the reviewer #1 for the feedback. Manuscript correction was performed.
Revised text: “…buccal cortical bone.” – Discussion section, page 6, line 390.
- A small comment about the role of PRF in a healing process would be recommended in the discussion.
Author’s response: We thank the reviewer #1 suggestion which objective was to enhance the quality of the manuscript. Following the reviewer’s recommendation, the role of L-PRF in the healing process was elaborated in the Discussion section.
Revised text: “L-PRF can be considered, more than a biomaterial, as a vital tissue precursor for tissue engineering. From the extracted blood, 97% of the platelets and more than 50% of the leukocytes are concentrated inside the PRF clot. After its application for approximately 2 weeks, growth factors such as PDGF, TGF-B1, VEGF and BMP-1 are released by platelets, which induce and control the proliferation and migration of regenerative cells, namely progenitor and stem cells. In addition, it acts as an immunomodulator and inflammation regulator [26,27].” - Discussion section, page 8, lines 608 to 614.
Reviewer 2 Report
Dear authors,
the article covers a very interesting topic.
Nevertheless I suggest some changes in order to improve the overall quality of the manuscript for the readers.
Line 78:
The authors wrote:
“After discussing treatment options (orthograde retreatment, apical microsurgery, IR and extraction)”
The authors should also add another treatment option: extraction with proper alveolus management and delayed implant placement.
Line 110:
The authors wrote:
“through sterile saline solution rinsing and periapical lesion and extruded filling material removal.”
The authors could explain better the procedures used to clean the alveolus after the extraction and before IR (instruments, etc.)
Figure 1: step “e”. please add a description of the alveolus management.
Introduction:
The authors could add a paragraph that could describe a possible alternative treatment plan that could be performed in similar clinical situations: orthodontic extrusion for implant site development.
The authors could therefore add a sentence like:
“Alternative treatment options could have involved guided eruption techniques. These have several benefits in dentistry. They can be used in several aspects such as tissue regeneration and implant site development either in lingual or conventional mechanics (you could cite Paolone MG, Kaitsas R, Paolone G, Kaitsas V. Lingual orthodontics and forced eruption: a means for osseous and tissue regeneration. Prog Orthod. 2008;9(2):46-57. PMID: 19350058.).
Line 277
The authors wrote:
“Regarding IR limitations, the inexistence of an established clinical protocol”
but at the same time in line 50 the authors wrote: “The IR technique has been progressively modified and refined”.
The two sentences may seem in contrast.
Author Response
- Dear authors, the article covers a very interesting topic. Nevertheless I suggest some changes in order to improve the overall quality of the manuscript for the readers.
Author’s response: We thank the reviewer #2 for the comments.
Revised text: Not applicable.
- Line 78: The authors wrote: “After discussing treatment options (orthograde retreatment, apical microsurgery, IR and extraction)”. The authors should also add another treatment option: extraction with proper alveolus management and delayed implant placement.
Author’s response: Once again we thank the reviewer #2 for the comment. The manuscript has been altered following the reviewer suggestion.
Revised text:
“(orthograde retreatment, apical microsurgery, IR and extraction with proper alveolus management and delayed implant placement)” – Case Report section, page 2, lines 83 to 85.
“(1) orthograde retreatment, (2) apical microsurgery, (3) IR or (4) extraction with proper alveolus management and delayed implant placement.” – Discussion section, page 6, lines 381 and 382.
- Line 110: The authors wrote: “through sterile saline solution rinsing and periapical lesion and extruded filling material removal.” The authors could explain better the procedures used to clean the alveolus after the extraction and before IR (instruments, etc.)
Author’s response: We thank the reviewer #2 for the comments.
Revised text: “While tooth’s extraoral handling procedures were being performed, the alveolus was prepared under 6x magnification. Alveolus preparation included a careful and meticulous curettage using a surgical curette for periapical lesion and extruded filling material removal, alternating with thorough sterile saline solution rinsing. The tooth was then replanted in the previously prepared alveolus (Figure 1m) and patient was instructed to bite on gauze.” – Case Report section, page 3, lines 228 to 231.
- Figure 1: step “e”. please add a description of the alveolus management.
Author’s response: We thank the reviewer #2 for the valuable feedback. However, the authors consider it would be more adequate to provide details concerning the alveolus management in Figure 1m.
Revised text: “Tooth replanted in the previously prepared alveolus (by a thorough sterile saline solution rinsing and periapical lesion and extruded filling material removal), without splinting.” – Case Report section, page 3, lines 207 and 208, Figure 1m.
- Introduction: The authors could add a paragraph that could describe a possible alternative treatment plan that could be performed in similar clinical situations: orthodontic extrusion for implant site development.The authors could therefore add a sentence like: “Alternative treatment options could have involved guided eruption techniques. These have several benefits in dentistry. They can be used in several aspects such as tissue regeneration and implant site development either in lingual or conventional mechanics (you could cite Paolone MG, Kaitsas R, Paolone G, Kaitsas V. Lingual orthodontics and forced eruption: a means for osseous and tissue regeneration. Prog Orthod. 2008;9(2):46-57. PMID: 19350058.).
Author’s response: We thank the reviewer #2 remarks which objective was to enhance the quality of the manuscript. Accordingly, the possibility of orthodontic extrusion has been addressed in the Discussion section.
Revised text: “Alternative treatment options could involve guided eruption techniques, as orthodontic extrusion 2 to 3 weeks prior to tooth extraction holds the potential of preventing intra- and post-operative IR complications [8]. This therapeutic approach can also be used for tissue regeneration and implant site development using either lingual or conventional mechanics [25].” – Discussion section, page 7, lines 439 to 443.
- Line 277: The authors wrote: “Regarding IR limitations, the inexistence of an established clinical protocol” but at the same time in line 50 the authors wrote: “The IR technique has been progressively modified and refined”. The two sentences may seem in contrast.
Author’s response: We thank the reviewer #2 for the risen point. However, although the intentional replantation procedure has been in fact progressively modified and refined over the years, a specific, recommended, step-by-step clinical protocol has not been established to date, with protocol variations being commonly found in different studies.
Revised text: Not applicable.
Reviewer 3 Report
The authors have managed a mandibular second molar with intentional replantation, followed by extraction, alveolar ridge preservation using L-PRF and implant-supported prosthetic rehabilitation. They have cited up-to-date references.
The figures are of high quality and comprehensive. However, I have the following comments that would make it better;
Title
The title is not concise and you need to cover other treatment steps not only replantation. You might consider the symptoms in the title
Abstract
"One year later, thermal and electrical tests remained 32 positive for tooth 36"
In the abstract, kindly delete this statement in order to focus on the treated tooth and avoid confusion
This treatment modality is know to be feasible. Therefore, the conclusion should be revised to show the novelty of this treatment option.
1. Introduction
L44
Replace "are" with "is"
L50-54
The sentence is lengthy. Kindly revise
L55
What do you mean by "perioperative"???
2. Case report
L66-67
Revise this sentence
L68
What do you mean by "coronary fissure"??
L94
What do you mean by 3.1?
L110
How did you confirm cleanliness of the alveolus before replantation?
L119
Delete "BOP"
L129
Delete "(20)"
L134
The citation location is not correct
L167
Delete "(6)"
L173
Replace "suture" with "suturing"
4. Discussion
L192
Add "s" to "examination"
L200
Add "option" after "alternative"
L202
Replace "bone cortical" with "cortical bone"
L213
Again, what do you mean with "perioperative"?
L214-215
Revise the sentence
L223
Delete "(MTA)"
L225
"A previous study...": make it as a new paragraph
L252
Delete "(IRM)"
5. Conclusions
This treatment modality is know to be feasible. Therefore, the conclusion should be revised to show the novelty of this treatment option.
Author Response
Dear Editor, we received the decision for "minor review" with reviewer 1 and 2 comments, plus academic editor comments, and prepared the revised version of the manuscript according to their demands. Right now, , I see comments from a reviewer third reviewer that are not answered point-by-point but which raise questions that seem to be answered by the present review, as follows:
Author’s Point-by-Point Response to Academic Editor and Reviewers
Academic Editor Comments
Dear authors, you made a great work! However, some improvements are mandatory before acceptance. 1) Abstract actually contains 300 words, the limit is about 200 words maximum. Please follow the style of structured abstracts: 1) Background; Methods; Results; 4) Conclusion. 2) All Figures, Schemes and Tables should be inserted into the main text close to their first citation and must be numbered following their number of appearance (Figure 1, Scheme I, Figure 2, Scheme II, Table 1, etc.). I suggest to arrange them in the text as suggested.
Author’s response: We want to thank the Editor for the opportunity of revising the manuscript and we carefully proceeded with the required modifications.
Reviewer #1
- The authors presented a case report of intentional replantation as a starting approach for a multidisciplinary treatment of a mandibular second molar. It is also a starting point for a discussion about possible treatment options in terms of overall success rate and long-term tooth prognosis observed in similar cases. The manuscript is well written with minor spell-check required. However there are some small remarks:
Author’s response: We thank the reviewer #1 for the comments.
Revised text: Not applicable.
- The diagnosis of symptomatic post-treatment apical periodontitis was established - maybe more accurate: chronic exacerbated apical periodontitis. How long ago the tooth was treated before? If there's a lesion it must have been a long time. Was it only a single treatment or multiple conservative approach? A bit more about history of the tooth would be recommended.
Author’s response: Following the reviewer #1 valuable suggestion, additional details regarding history of the tooth were added. Moreover, following the Glossary of Endodontic Terms provided by the American Association of Endodontists, pulp and periapical diagnosis nomenclature was updated.
Revised text:
“Previous medical history includes a non-surgical root canal treatment (NSRCT) perfomed in multiple sessions and involving an instrument fracture. The tooth kept asymptomatic for approximately 10 years after NSRCT.” – Case Report section, page 2, lines 70 to 73.
“Pulp diagnosis of previously treated and periapical diagnosis of symptomatic apical periodontitis were established.” – Case Report section, page 2, lines 82 and 83.
- Line 91: a diamond bur under copious irrigation… irrigation is more referred to endo procedure. Rather a water spray is visible on a figure.
Author’s response: Once again we thank the reviewer #1 for the risen point. The manuscript has been altered following the reviewer suggestion.
Revised text: “…under continuous water spray (Figure 1h)…” – Case Report section, page 3, line 222.
- Apicoectomy or apicectomy?
Author’s response: Both terms are synonymous and refer to the root-end resection procedure. However, according to the Glossary of Endodontic Terms provided by the American Association of Endodontists, apicoectomy is in fact the recommended term to use. Therefore, manuscript correction was carried out.
Revised text: “Apicoectomy (3 mm)…” – Case Report section, page 3, line 220.
- Shorter sentences are recommended eg: Line 134/135: rewrite. Confusing.
Author’s response: We thank the reviewer #1 for the comment. The highlighted sentence was rewritten.
Revised text: “At 2.5 years, radiographic examination of the replanted tooth showed extensive external root resorption [13]. Thus, clinical indication for tooth extraction was established (Figure 2c).” – Case Report section, page 4, lines 270 to 272.
- Amoxicillin with clavulanic acid: the dose and time shall also be added.
Author’s response: Following the reviewer’s instructions, details regarding antibiotic dosage were added.
Revised text: “…(amoxicillin and clavulanic acid 875/125 mg given every 12 hours for 8 days)…” – Case Report section, page 4, lines 261 and 262.
- Line 119: through ? or maybe „by a thorough saline solution rinsing”.
Author’s response: Following the reviewer’s suggestion, sentence was rewritten.
Revised text: “While tooth’s extraoral handling procedures were being performed, the alveolus was prepared under 6x magnification. Alveolus preparation included a careful and meticulous curettage using a surgical curette for periapical lesion and extruded filling material removal, alternating with thorough sterile saline solution rinsing. The tooth was then replanted in the previously prepared alveolus (Figure 1m) and patient was instructed to bite on gauze.” – Case Report section, page 3, lines 228 to 231.
- Line 205/206: Site preparation and implant placement (OsseoSpeedTM…) were carried out ( Fig..). The order.
Author’s response: Corrected.
Revised text: “…OsseoSpeedTM EV 5.4 S - 11 mm…” - Case Report section, page 5, lines 353 and 354.
- Line 259: buccal cortical bone.
Author’s response: We thank the reviewer #1 for the feedback. Manuscript correction was performed.
Revised text: “…buccal cortical bone.” – Discussion section, page 6, line 390.
- A small comment about the role of PRF in a healing process would be recommended in the discussion.
Author’s response: We thank the reviewer #1 suggestion which objective was to enhance the quality of the manuscript. Following the reviewer’s recommendation, the role of L-PRF in the healing process was elaborated in the Discussion section.
Revised text: “L-PRF can be considered, more than a biomaterial, as a vital tissue precursor for tissue engineering. From the extracted blood, 97% of the platelets and more than 50% of the leukocytes are concentrated inside the PRF clot. After its application for approximately 2 weeks, growth factors such as PDGF, TGF-B1, VEGF and BMP-1 are released by platelets, which induce and control the proliferation and migration of regenerative cells, namely progenitor and stem cells. In addition, it acts as an immunomodulator and inflammation regulator [26,27].” - Discussion section, page 8, lines 608 to 614.
Reviewer #2
- Dear authors, the article covers a very interesting topic. Nevertheless I suggest some changes in order to improve the overall quality of the manuscript for the readers.
Author’s response: We thank the reviewer #2 for the comments.
Revised text: Not applicable.
- Line 78: The authors wrote: “After discussing treatment options (orthograde retreatment, apical microsurgery, IR and extraction)”. The authors should also add another treatment option: extraction with proper alveolus management and delayed implant placement.
Author’s response: Once again we thank the reviewer #2 for the comment. The manuscript has been altered following the reviewer suggestion.
Revised text:
“(orthograde retreatment, apical microsurgery, IR and extraction with proper alveolus management and delayed implant placement)” – Case Report section, page 2, lines 83 to 85.
“(1) orthograde retreatment, (2) apical microsurgery, (3) IR or (4) extraction with proper alveolus management and delayed implant placement.” – Discussion section, page 6, lines 381 and 382.
- Line 110: The authors wrote: “through sterile saline solution rinsing and periapical lesion and extruded filling material removal.” The authors could explain better the procedures used to clean the alveolus after the extraction and before IR (instruments, etc.)
Author’s response: We thank the reviewer #2 for the comments.
Revised text: “While tooth’s extraoral handling procedures were being performed, the alveolus was prepared under 6x magnification. Alveolus preparation included a careful and meticulous curettage using a surgical curette for periapical lesion and extruded filling material removal, alternating with thorough sterile saline solution rinsing. The tooth was then replanted in the previously prepared alveolus (Figure 1m) and patient was instructed to bite on gauze.” – Case Report section, page 3, lines 228 to 231.
- Figure 1: step “e”. please add a description of the alveolus management.
Author’s response: We thank the reviewer #2 for the valuable feedback. However, the authors consider it would be more adequate to provide details concerning the alveolus management in Figure 1m.
Revised text: “Tooth replanted in the previously prepared alveolus (by a thorough sterile saline solution rinsing and periapical lesion and extruded filling material removal), without splinting.” – Case Report section, page 3, lines 207 and 208, Figure 1m.
- Introduction: The authors could add a paragraph that could describe a possible alternative treatment plan that could be performed in similar clinical situations: orthodontic extrusion for implant site development.The authors could therefore add a sentence like: “Alternative treatment options could have involved guided eruption techniques. These have several benefits in dentistry. They can be used in several aspects such as tissue regeneration and implant site development either in lingual or conventional mechanics (you could cite Paolone MG, Kaitsas R, Paolone G, Kaitsas V. Lingual orthodontics and forced eruption: a means for osseous and tissue regeneration. Prog Orthod. 2008;9(2):46-57. PMID: 19350058.).
Author’s response: We thank the reviewer #2 remarks which objective was to enhance the quality of the manuscript. Accordingly, the possibility of orthodontic extrusion has been addressed in the Discussion section.
Revised text: “Alternative treatment options could involve guided eruption techniques, as orthodontic extrusion 2 to 3 weeks prior to tooth extraction holds the potential of preventing intra- and post-operative IR complications [8]. This therapeutic approach can also be used for tissue regeneration and implant site development using either lingual or conventional mechanics [25].” – Discussion section, page 7, lines 439 to 443.
- Line 277: The authors wrote: “Regarding IR limitations, the inexistence of an established clinical protocol” but at the same time in line 50 the authors wrote: “The IR technique has been progressively modified and refined”. The two sentences may seem in contrast.
Author’s response: We thank the reviewer #2 for the risen point. However, although the intentional replantation procedure has been in fact progressively modified and refined over the years, a specific, recommended, step-by-step clinical protocol has not been established to date, with protocol variations being commonly found in different studies.
Revised text: Not applicable.
Round 2
Reviewer 3 Report
My comments have not been addressed
Author Response
We thank reviewer #3 for the valuable comments and time spent to assess the manuscript. We appreciate your insight to improve the quality of this manuscript.
Reviewer #3
The authors have managed a mandibular second molar with intentional replantation, followed by extraction, alveolar ridge preservation using L-PRF and implant-supported prosthetic rehabilitation. They have cited up-to-date references.
The figures are of high quality and comprehensive. However, I have the following comments that would make it better;
Author’s response: We thank the reviewer #3 for the comments.
Revised text: Not applicable.
Title
The title is not concise and you need to cover other treatment steps not only replantation. You might consider the symptoms in the title
Author’s response: We thank the reviewer #3 for the risen point. However, we want to have a focus on the primary intervention and the need for a multidisciplinary approach to deliver high quality patient care.
Revised text: Not applicable.
Abstract
"One year later, thermal and electrical tests remained positive for tooth 36"
In the abstract, kindly delete this statement in order to focus on the treated tooth and avoid confusion
Author’s response: We thank the reviewer #3 for the risen point. As suggested, we removed this sentence.
Revised text: Not applicable.
This treatment modality is know to be feasible. Therefore, the conclusion should be revised to show the novelty of this treatment option.
Author’s response: We thank the reviewer #3 for the risen point. However, this is a case report and our main concern is to present balanced and evidence-based conclusions, ground on the best treatment approaches currently available and implemented in the treatment of this particular patient.
- Introduction
L44
Replace "are" with "is"
Author’s response: Corrected.
L50-54
The sentence is lengthy. Kindly revise
Author’s response: We consider the sentence correct and comprehensible.
L55
What do you mean by "perioperative"???
Author’s response: According to a Medical Dicionary it means “Around the time of surgery. This usually lasts from the time the patient goes into the hospital or doctor's office for surgery until the time the patient goes home.” Another option would be intraoperative, but in our opinion perioperative allows a better focus on the aspects that may influence the outcome.
- Case report
L66-67
Revise this sentence
Author’s response: We consider the sentence correct and comprehensible.
L68
What do you mean by "coronary fissure"??
Author’s response: We thank the reviewer #3 for the risen point. We think this can be better described as a “coronal vertical craze line”.
Revised text: “…a distal coronal vertical craze line and increased probing pocket depth…” Line 76.
L94
What do you mean by 3.1?
Author’s response: We thank the reviewer #3 for the risen point. The subsections and sections numeration has been corrected.
L110
How did you confirm cleanliness of the alveolus before replantation?
Author’s response: This has been corrected to satisfy reviewer’s #1 concern 7.
Revised text: “While tooth’s extraoral handling procedures were being performed, the alveolus was prepared under 6x magnification. Alveolus preparation included a careful and meticulous curettage using a surgical curette for periapical lesion and extruded filling material removal, alternating with thorough sterile saline solution rinsing. The tooth was then replanted in the previously prepared alveolus (Figure 1m) and patient was instructed to bite on gauze.” – Case Report section, page 3, lines 228 to 231.
L119
Delete "BOP"
Author’s response: We thank the reviewer #3 for the risen point. We deleted as requested.
L129
Delete "(20)"
Author’s response: We thank the reviewer #3 for the risen point. We deleted as requested.
L134
The citation location is not correct
Author’s response: We thank the reviewer #3 for the risen point. There are a considerable number of different diagnostic terms for this clinical situation, that’s why we think it is important to cite this reference at this point.
L167
Delete "(6)"
Author’s response: We thank the reviewer #3 for the risen point. We deleted as requested.
L173
Replace "suture" with "suturing"
Author’s response: We thank the reviewer #3 for the risen point. We changed accordingly to the suggestion.
- Discussion
L192
Add "s" to "examination"
Author’s response: We thank the reviewer #3 for the risen point. We changed accordingly to the suggestion.
L200
Add "option" after "alternative"
Author’s response: We thank the reviewer #3 for the risen point. We changed accordingly to the suggestion.
L202
Replace "bone cortical" with "cortical bone"
Author’s response: We thank the reviewer #3 for the risen point. We changed accordingly to the suggestion.
L213
Again, what do you mean with "perioperative"?
Author’s response: According to a Medical Dicionary it means “Around the time of surgery. This usually lasts from the time the patient goes into the hospital or doctor's office for surgery until the time the patient goes home.” Another option would be intraoperative, but in our opinion perioperative allows a better focus on the aspects that may influence the outcome.
L214-215
Revise the sentence
Author’s response: We thank the reviewer #3 for the risen point.
Revised text: “…The number of periodontal pockets (≥ 6mm) and patient age (> 40 years old) compose key pre-operative aspects that may jeopardize the success of IR procedure …” Line 425-27.
L223
Delete "(MTA)"
Author’s response: We thank the reviewer #3 for the risen point. We deleted as requested.
L225
"A previous study...": make it as a new paragraph
Author’s response: We thank the reviewer #3 for the risen point. We changed accordingly to the suggestion.
L252
Delete "(IRM)"
Author’s response: We thank the reviewer #3 for the risen point. We deleted as requested.
- Conclusions
This treatment modality is know to be feasible. Therefore, the conclusion should be revised to show the novelty of this treatment option.
Author’s response: We thank the reviewer #3 for the risen point. However, this is a case report and our main concern is to present balanced and evidence-based conclusions, ground on the best treatment approaches currently available and implemented in the treatment of this particular patient.